# Mitochondrial Uncoupling Proteins (UCP1-UCP3) and Adenine Nucleotide Translocase (ANT1) Enhance the Protonophoric Action of 2,4-Dinitrophenol in Mitochondria and Planar Bilayer Membranes

**DOI:** 10.3390/biom11081178

**Published:** 2021-08-09

**Authors:** Kristina Žuna, Olga Jovanović, Ljudmila S. Khailova, Sanja Škulj, Zlatko Brkljača, Jürgen Kreiter, Elena A. Kotova, Mario Vazdar, Yuri N. Antonenko, Elena E. Pohl

**Affiliations:** 1Institute of Physiology, Pathophysiology and Biophysics, University of Veterinary Medicine, A-1210 Vienna, Austria; kristina.zuna@vetmeduni.ac.at (K.Ž.); olga.jovanovic@vetmeduni.ac.at (O.J.); jürgen.kreiter@vetmeduni.ac.at (J.K.); 2Belozersky Institute of Physico-Chemical Biology, Lomonosov Moscow State University, Leninskie Gory 1/40, 119991 Moscow, Russia; khailova@genebee.msu.ru (L.S.K.); kotova@belozersky.msu.ru (E.A.K.); 3Department of Chemistry, Faculty of Science, University of Zagreb, Horvatovac 102a, 10000 Zagreb, Croatia; Sanja.Skulj@irb.hr; 4Division of Organic Chemistry and Biochemistry, Ruđer Bošković Institute, Bijenička 54, 10000 Zagreb, Croatia; Zlatko.Brkljaca@irb.hr (Z.B.); Mario.Vazdar@irb.hr (M.V.); 5Institute of Organic Chemistry and Biochemistry, Czech Academy of Sciences, Flemingovo nám. 2, 16610 Prague, Czech Republic

**Keywords:** mitochondrial uncoupler, protonophore, membrane potential, proton conductance, artificial membranes, molecular dynamics simulations

## Abstract

2,4-Dinitrophenol (DNP) is a classic uncoupler of oxidative phosphorylation in mitochondria which is still used in “diet pills”, despite its high toxicity and lack of antidotes. DNP increases the proton current through pure lipid membranes, similar to other chemical uncouplers. However, the molecular mechanism of its action in the mitochondria is far from being understood. The sensitivity of DNP’s uncoupling action in mitochondria to carboxyatractyloside, a specific inhibitor of adenine nucleotide translocase (ANT), suggests the involvement of ANT and probably other mitochondrial proton-transporting proteins in the DNP’s protonophoric activity. To test this hypothesis, we investigated the contribution of recombinant ANT1 and the uncoupling proteins UCP1-UCP3 to DNP-mediated proton leakage using the well-defined model of planar bilayer lipid membranes. All four proteins significantly enhanced the protonophoric effect of DNP. Notably, only long-chain free fatty acids were previously shown to be co-factors of UCPs and ANT1. Using site-directed mutagenesis and molecular dynamics simulations, we showed that arginine 79 of ANT1 is crucial for the DNP-mediated increase of membrane conductance, implying that this amino acid participates in DNP binding to ANT1.

## 1. Introduction

Obesity is one of the most acute health problems in developed countries worldwide. Recent statistics from the World Health Organization show that obesity has nearly tripled worldwide since 1975 (https://www.who.int/news-room/fact-sheets/detail/obesity-and-overweight, accessed on 5 June 2021). Although the uncoupling of oxidative phosphorylation (OxPhos) in the mitochondria was long ago proposed as a target for treating obesity, no breakthrough therapy has yet been established. 2,4-Dinitrophenol (DNP) has been known since the mid-1930s as an effective component of “diet pills” that is capable of reducing obesity by increasing the basal metabolic rate [1]. Severe side effects forced its withdrawal from the pharmaceutical market [2,3]. However, DNP can still be illegally purchased under different names over the internet as a drug conferring rapid and supposedly safe weight loss, and is also used as an illegal food supplement [4].

Sustained scientific interest in DNP is based not only on its much higher efficacy than other drugs but also on the need to develop antidotes against its toxicity to prevent the increasing number of deaths associated with its uncontrolled use. The detailed molecular mechanism of the uncouplers’ action in the mitochondria remains unclear, which hinders the use of DNP and its derivatives as therapeutic drugs.

Since DNP increases the conductance of pure lipid membranes, similar to other uncouplers [5,6,7,8,9,10,11,12,13,14], its effect on the mitochondria was ascribed to an increase in proton permeability through the lipid part of the inner mitochondrial membrane (IMM). On the other hand, the azido-substituted analog of DNP, 2-azido-4-nitrophenol, was shown to bind effectively to mitochondrial proteins [15,16,17]. In addition, the uncoupling effect of DNP on the mitochondria was shown to be partially blocked by carboxyatractyloside (CATR) [18,19], a specific inhibitor of the adenine nucleotide translocase (ANT), indicating the participation of ANT in the uncoupling action of DNP in the mitochondria.

Partial suppression of the uncoupling effect was also found for other protonophores, including BAM15 [19,20,21,22,23,24]. These findings support the hypothesis of ANT’s participation in the uncoupling action of certain protonophores in the mitochondria [25].

Our previous study demonstrated the ability of ANT1 to enhance the proton current across artificial membranes in the presence of weak uncouplers—long-chain fatty acids (FA) [26]. We now hypothesize that DNP’s protonophoric function may be also potentiated by mitochondrial proton-transporting proteins, in addition to the DNP-mediated increase in the selective lipid membrane’s permeability for protons. To test this hypothesis, we used a well-defined model of lipid bilayer membranes reconstituted with recombinant ANT1 or UCPs (1) to investigate the contribution of these proteins to DNP-mediated proton leakage, (2) to evaluate the efficiency of the inhibitors of proton transport mediated by ANT1/UCP, and (3) to reveal the amino acids crucial for ANT1 activation by DNP.

## 2. Materials and Methods

### 2.1. Chemicals

2,4-Dinitrophenol (DNP), chloroform, dimethyl sulfoxide (DMSO), 1,2-dioleoyl-sn-glycero-3-phosphotidylcholine (DOPC), 1,2-dioleoyl-sn-glycero-3-phosphoethanolamine (DOPE), cardiolipin (CL), sodium sulfate (Na_2_SO_4_), 2-(*N*-morpholino) ethanesulfonic acid (MES), tris(hydroxymethyl)-aminomethane (Tris), adenosine 5′-triphosphate (ATP), guanosine 5′-triphosphate (GTP), carboxyatractyloside (CATR), ethylenediaminetetraacetic acid (EDTA), ethylene glycol-bis(β-aminoethyl ether)-N,N,N′,N′-tetraacetic acid (EGTA), *N*-lauroylsarcosine, Triton X-114, dithiothreitol (DTT), bovine serum albumin (BSA), sucrose, 3-(*N*-morpholino) propanesulfonic acid (MOPS), rotenone, and safranin O were purchased from Sigma-Aldrich (Vienna, Austria). Arachidonic acid (AA) was purchased from Larodan (Solna, Sweden), n-octylpolyoxyethylene was purchased from BACHEM (Bubendorf, Switzerland), and hydroxyapatite was purchased from Bio-Rad Laboratories (Hercules, CA, USA).

### 2.2. Cloning, Purification, and Reconstitution of Murine ANT1 and UCPs

Cloning, purification, and reconstitution of murine ANT1, UCP1, UCP2, and UCP3 followed previously established protocols [27,28,29]. In brief, expression plasmids containing selected cDNA sequences were transformed into the *E. coli* Rosetta DE3 strain (Novagen (Merck), Darmstadt, Germany). After induction, high-pressure homogenization, and centrifugation, the proteins were isolated as inclusion bodies. For protein refolding, purification, and reconstitution, 1 mg of the inclusion bodies was solubilized in a TE/G buffer containing 2% *N*-lauroylsarcosine, 1.3% Triton X-114, 0.3% n-octylpolyoxyethylene, 1 mM DTT, and GTP at pH 7.5. Fifty milligrams of the lipid mixture (DOPC:DOPE:CL; 45:45:10 mol%) was mixed in gradually. The mixture was concentrated and dialyzed against a buffer used in the experiments (50 mM Na_2_SO_4_, 10 mM MES, 10 mM Tris, and 0.6 mM EGTA at pH 7.34). Unfolded and aggregated proteins were removed from the dialysate by centrifugation and through a column containing hydroxyapatite; non-ionic detergents were removed by applying Bio-Beads SM-2 (Bio-Rad Laboratories, Hercules, CA, USA). The protein concentration in proteoliposomes was measured with the Micro BCA Protein Assay Kit (Thermo Fisher Scientific, Waltham, MA, USA). Protein purity was verified by SDS-PAGE and silver staining (Appendix A). Proteoliposomes were produced in independent batches. The following batch numbers were used for this study: ANT1 #42 and #47, UCP1 #117, UCP2 #39, UCP3 #31, and ANT1 R79S #3.

### 2.3. Generation of the ANT1 R79S Mutant

In vitro site-directed mutagenesis was carried out on expression plasmids containing the cDNA of mANT1 as a template. The mutation was introduced with an oligonucleotide designed to alter codon Arg79 (CGG) to Ser (TCG) using a QuikChange II site-directed mutagenesis kit (Agilent Technologies, Vienna, Austria). The successful introduction of mutations was confirmed by sequencing. Mutant ANT1 expression plasmids were transformed in the *E. coli* expression strain Rosetta DE3. Expression induction, inclusion body isolation, and reconstitution into liposomes of the ANT1 mutant were performed as described above for the ANT1 wild-type.

### 2.4. Formation of Planar Lipid Bilayer Membranes and Membrane Conductance Measurements

Planar lipid bilayers were formed from proteoliposomes as described previously [30,31]. Correct membrane formation was verified by measuring the membrane capacitance (C = 0.72 ± 0.02 μF/cm^2^), which was independent of the presence of protein, AA and DNP. Current voltage measurements were performed with a patch-clamp amplifier (EPC 10, HEKA Elektronik, Dr. Schulze GmbH, Lambrecht, Germany). The specific total membrane conductance (G_m_) at 0 mV was obtained as the slope of a linear fit of the experimental data at the applied voltages from −50 mV to +50 mV, normalized to the membrane area. AA solved in chloroform was added to the membrane-forming lipid solution. ATP dissolved in the assay buffer (pH = 7.34), CATR, and DNP (both dissolved in DMSO) were added to the buffer solution before forming bilayer membranes. Control experiments showed that DMSO did not change the total membrane conductance of the lipid bilayers in the range of concentrations used [32]. The concentrations of each substrate are indicated in the figure legends. The relative conductance was calculated according to Equation (1):(1)$$G_{rel}=\frac{G-G_{0}}{G_{1}-G_{0}}$$
where *G*_0_ is the total membrane conductance of lipid membranes reconstituted with DNP, *G*_1_ is the total membrane conductance of lipid membranes reconstituted with ANT1 and DNP, and *G* is the total specific membrane conductance of lipid membranes reconstituted with ANT1, DNP, and/or CATR/ATP (Figure 3).

### 2.5. Isolation of Rat Liver Mitochondria

Rat liver or heart mitochondria (RLM, RHM) were isolated by differential centrifugation [33] in a medium containing 250 mM sucrose, 5 mM MOPS, 1 mM EGTA, and bovine serum albumin (0.5 mg/mL) at pH 7.4. The final washing was performed in a medium with the same composition. Protein concentration was determined using the Biuret method. Handling of animals and experimental procedures were conducted in accordance with international guidelines for animal care and use and were approved by the Ethics Committee of the Belozersky Institute of Physico-Chemical Biology at Moscow State University (protocol #3 from 12 February 2018).

### 2.6. Membrane Potential (Δψ) Measurements in Isolated Mitochondria

The transmembrane electric potential difference (Δψ) was measured using safranin O dye [34]. An incubation medium containing 250 mM sucrose, 5 mM MOPS, 1 mM EGTA, 2 μM rotenone, 5 mM succinate (pH 7.4), and 15 μM safranin O was used. The mitochondrial protein content measured by the Biuret method was 0.6 mg protein/mL. The experiments were carried out at 26 °C.

The difference in absorbance between 555 and 523 nm (Δ*A*) was recorded with an Aminco DW-2000 spectrophotometer (Olis Inc., Bogart, GA, USA) in dual wavelength mode.

Mitochondrial membrane potential (MMP) was normalized according to Equation (2):(2)$$\mathrm{Absorbance},\;\%\;\mathrm{of}\;\mathrm{the}\;\mathrm{maximum}=\frac{\Delta A_{max}-\Delta A}{\Delta A_{max}-\Delta A_{min}}\times 100$$
where Δ*A* is the difference between A_523_ and A_555_, Δ*A_max_* is the highest membrane potential measured without the presence of DNP, and Δ*A_min_* is the lowest MMP measured in the presence of DNP at the highest concentration. Statistical analysis was performed using a Student’s test with a level of significance of 0.01.

### 2.7. Molecular Dynamic Simulations

We performed all-atom molecular dynamic (MD) simulations of wild-type and mutated (R79 to S79) ANT1 protein in a 1,2-dioleoyl-sn-glycero-3-phosphocholine (DOPC) bilayer. The initial structure for both the wild-type and mutated forms of ANT1 was taken from the end of the 2 μs simulation of the wild-type ANT1 protein in the DOPC bilayer [35], with the mutation of R79 into S79 and the subsequent introduction of the mutated form of ANT1 into the DOPC bilayer being performed using CHARMM-GUI (http://www.charmm-gui.org/, accessed on 10 February 2021) [36,37,38]. Overall, 4 system set-ups were prepared: the wild-type ANT1 with the anionic form of 2,4-dinitrophenol (DNP) (i) or ATP^4−^ bound in the cytosolic-open state (ii), as well as the mutated ANT1 with the DNP anion (iii) or ATP^4−^ bound in the same position (iv). Since DNP has a p*K*a value in water of 4.1, we assumed it was anionic at the neutral pH used in MD simulations. All simulation boxes contained ANT1 protein (wild-type or the R79-to-S79 mutant) (with a total charge of +19e or +18e, respectively), 73 DOPC molecules per leaflet (146 per system), 11,500 water molecules, a single DNP anion or ATP^4−^, and the necessary number of Cl^−^ anions to neutralize the net charge, depending on whether wild-type ANT1 or its mutant was present in the system. ANT1 protein, ATP^4−^ ion, and DOPC lipids were described by the CHARMM36m force field [39]. The force field for the anionic form of the DNP (2,4-dinitrophenol) anion was built on the basis of the CHARMM general force field (CGenFF) [40] (https://cgenff.umaryland.edu/, accessed on 10 February 2021).

The system containing mutated ANT1 protein, DOPC molecules, and water/ions (no DNP anion or ATP^4−^ present at this stage) was first minimized and equilibrated in 6 steps using the CHARMM-GUI protocol [41] and then simulated for a further 500 ns without any restraints with a 2 fs time step in a periodic rectangular box of 7.9 nm × 7.9 nm × 9.4 nm using the isobaric–isothermal ensemble (NPT) and periodic boundary conditions in all directions at T = 310 K, maintained via a Nosé–Hoover thermostat [42] independently for the DOPC, water/ions, and protein subsystems, with a coupling constant of 1.0 ps^−1^. The pressure was set to 1.013 bar and controlled with a semi-isotropic Parrinello–Rahman barostat [43], with a time constant for pressure coupling of 5 ps^−1^. Long-range electrostatics were calculated using the particle-mesh Ewald (PME) method [44] with real-space Coulomb interactions cut off at 1.2 nm using a Fourier spacing of 0.12 nm and a Verlet cut-off scheme.

To prepare the investigated systems, one molecule of DNP anion (or ATP^4−^) was placed in the cytosolic-open state of the equilibrated systems containing wild-type ANT1 (obtained after 2 μs of free MD simulation) or mutated ANT1 (see previous paragraph). More precisely, the DNP anion (or ATP^4−^) was placed in the cavity of the ANT1 protein, in proximity to the R79 (wild-type) or S79 (mutated ANT1) amino acid residue, using the VMD molecular graphics program [45]. The prepared systems were then minimized, and short preliminary simulations (2 ns) with positional restraints on the DNP (or ATP) molecule (500 kJ mol^−1^ nm^−2^) were conducted to relax the protein around the DNP molecule. Subsequently, both investigated systems were propagated for a duration of 500 ns, with MD parameters equal to those used to propagate the mutated ANT1 system (see previous paragraph). All simulations were run with the GROMACS 2018 software package [46].

### 2.8. Statistics

Data analysis and fitting of electrophysiological measurements were performed using Sigma Plot 12.5 (Systat Software GmbH, Erkrath, Germany) and are displayed as the mean ± SD of at least 3 independent experiments. Each independent experiment involved at least 3 measurements using independently formed bilayer membranes.

## 3. Results

### 3.1. CATR Recouples Mitochondria Uncoupled by DNP

Measurements of oxygen consumption in isolated rat liver mitochondria (RLM) and skeletal muscle mitochondria showed that CATR can partially reverse the uncoupling effect of low DNP concentrations [18,47]. Because mitochondrial membrane potential (MMP) is a more sensitive indicator of mitochondrial coupling, we measured the MMP of isolated RLM in the presence of DNP and CATR using the potential-sensitive dye safranin O [34]. Figure 1A shows that, after the addition of 10–50 µM DNP to RLM, MMP decreased in a concentration-dependent manner. CATR at a concentration of 3 µM partially restored MMP at all concentrations, supporting the data obtained from oxygen consumption measurements earlier.

Because the DNP effect was small in RLM, we performed similar experiments with mitochondria isolated from rat hearts (RHM), which are known to have a higher ANT1 abundance [48]. The heart is also known to express another proton-transporting protein, UCP3 [49,50]. Figure 1B shows that the uncoupling effect of DNP in RHM was significantly higher than that in RLM. Importantly, the subsequent recoupling effect of CATR was also much more pronounced in RHM. A significant difference (*p* < 0.01) was found between RLM and RHM for both DNP-mediated uncoupling and recoupling in the presence of CATR (Figure 1C). Nonetheless, no complete recoupling was observed, suggesting a significant contribution of the DNP-assisted proton shuttling through the lipid part of membranes to the overall DNP-mediated proton transport and/or the involvement of other carrier proteins of IMM in this process.

### 3.2. DNP Increases the Proton Conductance of the Membranes Reconstituted with Mitochondrial Membrane Proteins

It was previously shown that ANT1 mediates proton transport in the presence of free long-chain fatty acids (FA), similar to UCPs [26,47,51]. To test whether DNP also activates ANT1, we used a well-defined model of planar bilayer membranes reconstituted with recombinant murine ANT1 (Appendix A) [27]. In contrast to the well-known Müller–Rudin technique, in which the membrane is “painted” by lipids dissolved in n-decane, we folded bilayers from the solvent-free monolayer formed from the proteoliposomes. The specific conductance (G_m_) of the membranes reconstituted with ANT1 (G_m_ = 11.1 ± 0.8 nS/cm^2^) was comparable with the conductance of pure lipid membranes made from DOPC, DOPE, and cardiolipin (CL) (45:45:10 mol%; G_m_ = 10.81 ± 2.72 nS/cm^2^, Figure 2A).

Figure 2A shows that the specific conductance of membranes containing DNP in the presence of ANT1 (ANT1 + DNP, red column; G_m_ = 80.9 ± 9.4 nS/cm^2^) was approximately double that of membranes containing only DNP at a concentration of 50 µM (DNP, gray column; G_m_ = 42.7 ± 5.9 nS/cm^2^). Until now, only FAs have been shown to activate ANT1 directly (Figure 2A) [18,26,51,52].

To test whether uncoupling proteins can be activated by DNP in the absence of FA, we reconstituted recombinant murine UCP1, UCP2, and UCP3 (Appendix A) in bilayer membranes made of DOPC, DOPE, and CL. G_m_ in the presence of UCPs and DNP (G_m, UCP1_ = 67.1 ± 7.2 nS/cm^2^, G_m, UCP2_ = 85.1 ± 8.4 nS/cm^2^; G_m, UCP3_ = 72.1 ± 9.1 nS/cm^2^) was again approximately double that in the presence of DNP alone (G_m_ = 42.7 ± 5.9 nS/cm^2^)(Figure 2B).

The contribution of proton conductance to the total membrane conductance (G_H/OH_/G_0_) was determined from the shift in reverse potential in the presence of a transmembrane pH gradient of 0.4 [30]. Figure 2C shows that in the presence of DNP and protein (ANT1 or UCP1), the G_H/OH_/G_0_ was approximately 1, confirming the specificity of the proton transport in all cases.

### 3.3. CATR and ATP Inhibit DNP-Mediated Activation of ANT1 Only If Added before DNP

Because CATR was able to partially restore MMP in mitochondria treated with DNP (Figure 1), we tested whether its addition would deplete the proton transport mediated by DNP in the presence of reconstituted ANT1. Interestingly, CATR decreased G_m_ more effectively if added prior to DNP (Figure 3A). The decrease was more pronounced at the higher CATR concentration of 100 µM (~98%), indicating the specificity of the interaction.

ATP was shown to be an inhibitor of FA-activated ANT1 [26], so we examined its effect on DNP-mediated proton transport as well. Figure 3B shows that 4 mM ATP inhibited the effect of DNP by ~96%, further suggesting the same putative binding site for DNP, CATR, ATP, and FA. The total membrane conductance (G_m_) values of Figure 3 are shown in Appendix A. ATP inhibited the UCP1–UCP3-enhanced DNP-mediated uncoupling in the same manner as it did in case of ANT1 (Figure 4).

### 3.4. R79 Is Crucial for DNP–Protein Interaction

Because arginine 79 (R79) was previously shown to be important for the binding of CATR and ATP to ANT1 [53,54,55], we further evaluated the role of R79 in the interaction between DNP and ANT1. Indeed, the recombinant ANT1 in which R79 was substituted by serine using site-directed mutagenesis (ANT1 R79S) was unable to facilitate proton transport mediated by DNP (Figure 5, dark red columns). This result supported the idea that the R79 of ANT1 is crucial for the increase in proton transport in the presence of DNP.

### 3.5. Molecular Dynamic Simulations Show the Binding of DNP to R79 but Not to S79

To test the putative role of R79 in the interaction of DNP with ANT1, we performed 500 ns molecular dynamic (MD) simulations for two systems: ANT1 and ANT1 R79S. Figure 6 shows that DNP bound easily to R79 in ANT1, whereas in the case of ANT1 R79S the binding was absent. Analysis of the distances between DNP and the C_Z_ atom of R79 (Figure 6A) shows that the binding of DNP to R79 was not very strong and persistent, as evidenced by the relatively large corresponding average distance (Figure 6B). However, DNP frequently makes close contact with R79 by forming bidentate hydrogen bonds with its guanidinium side chain, which is apparently responsible for DNP binding in addition to attractive electrostatic interactions (Figure 6B, arrows; Figure 6C). In contrast, a mutation of R79 to S79 in ANT1 abolished the close contacts between DNP and the C_B_ atom of serine due to the loss of attractive electrostatic interactions and favorable hydrogen bonding (Figure 6D).

However, the binding of DNP to R79 was not as strong as in the case of ATP [56,57]. Appendix A presents an analysis of the corresponding distances between phosphorous atoms in the phosphate groups of ATP (Pα, Pβ, and Pγ) and the C_Z_ atom in R79 in ANT1 (Appendix A) and the C_B_ atom in S79 in the ANT1 R79S (Appendix A), respectively. We can see that ATP bound very strongly to R79, as evidenced by the short and persistent contacts between Pα and Pβ atoms and R79 throughout the MD simulations. However, when R79 was mutated, the average distances from all three phosphate groups to S79 increased, showing weaker binding to S79 due to the loss of electrostatic interactions.

## 4. Discussion

Comparing the conductance of the pure lipid bilayer membranes and membranes reconstituted with recombinant ANT1, UCP1, UCP2, or UCP3, we revealed that proteins potentiate DNP-mediated proton transport through the membrane. Until now, only FAs were known to serve as a co-factor of the uncoupling function of these proteins. The mutation R79 in ANT1 abolished the increment in the conductance of the bilayer membrane reconstituted with protein, thus indicating DNP’s putative binding site.

Our experiments on isolated mitochondria and membranes reconstituted with ANT1 or UCPs showed that DNP at a concentration of 50 µM can significantly decrease the MMP (Figure 1) and increase G_m_ (Figure 2). This is in agreement with a previous report about the large effects of 30 µM DNP on the mitochondria [58]. Interestingly, the fact that IMM has very high protein to lipid ratio of approximately 3:1 implies that the interaction of DNP with a protein contributes significantly to the DNP’s action on the mitochondria.

Both CATR and ATP were able to prevent the increase in the protonophoric action of DNP in the presence of protein, which suggested the involvement of a similar binding site. MD simulation results supported the idea that ATP binds to R79 more strongly than DNP, showing much shorter distances between ATP phosphate groups and R79, and persistent contacts between them (Appendix A). DNP also bound to R79 but, based on the distance analysis (Figure 6B), the binding of DNP was not as strong, and close contacts between DNP and R79 are not as frequent as in ATP binding.

The results of the site-directed mutagenesis showed that R79 is crucial for the DNP effect, since its replacement completely abolished the G_m_ increase in the presence of protein. R79 is a component of the arginine ring made of three arginine residues, which was found to protrude into the center of the protein’s cavity [55]. At the alkaline pH of the cytosol and mitochondrial matrix, which we imitated in our experiments, the aforementioned molecules possess at least one negative charge and interact with positively charged amino acids, with R79 either being one of them or being the critical one. As R79 plays an important role in the binding of all ANT1-specific substrates, it is possibly a highly conserved residue for the protein’s functions—ATP/ADP and proton transport—regardless of which molecule acts as an uncoupler. This was also shown by the MD simulations, suggesting that the mutation of R79 to S79 weakens the interaction between DNP and the protein due to the loss of electrostatic attraction and hydrogen bonding possibilities (Figure 6C,D).

We suggest that the mechanism of the enhancement of DNP’s protonophoric function by ANT1 is similar to that of fatty acids. So far, two hypotheses have been proposed to explain the induction of proton conductance by fatty acids involving ANT and uncoupling proteins (for a review, see [59]): (1) Protein accelerates the transport of the FA anion, which limits total proton transport by FA flip-flop through the lipid membrane [26,60]; (2) FA induces a conformational transition in protein, leading to proton leakage through the protein [51,61]. The present work argued in favor of the first mechanism of the induction of proton conductance by ANT1 [26] because FA and DNP share the R79 residue as a common binding site [62]. Further experiments investigating the exact interaction between ANT1 and DNP and outlining the mechanism are required.

Besides ANT1, UCP1–UCP3 enhance the proton transport mediated by DNP (Figure 2B). This is in agreement with the experiments on isolated rat heart mitochondria, in which we observed a more pronounced effect of DNP than in liver. Heart mitochondria not only have a higher level of ANT1 than RLM [48] but also express UCP3, which is not found in the liver [49,63,64]. Notably, the R79 of ANT1 is highly conserved among the SLC25 protein family, with the residues R84, R88, and R84 being homologs of it in UCP1, UCP2, and UCP3, respectively. These residues likely form a putative DNP binding site in uncoupling proteins, and their depletion would result in the complete loss of DNP-mediated proton leakage, as occurred in R79S ANT1. Furthermore, we suggest that other proton-transporting proteins, such as the aspartate/glutamate carrier, the dicarboxylate carrier, the phosphate carrier, and the 2-oxoglutarate carrier [65,66,67,68], will also potentiate the protonophoric function of DNP.

## Figures and Tables

**Figure 1 biomolecules-11-01178-f001:**
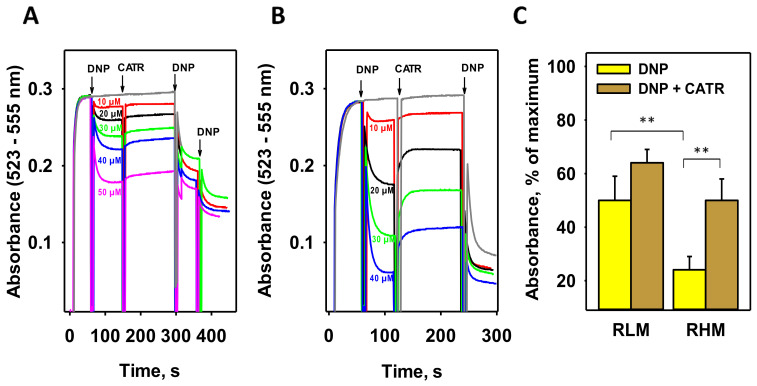
CATR recovers the DNP-mediated decrease in mitochondrial membrane potential (MMP) in isolated rat mitochondria. Representative recordings of MMP in rat liver mitochondria (**A**) and heart (**B**) mitochondria correspond to changes in the absorbance of the potential-sensitive dye safranin O (15 µM). CATR was added at a final concentration of 3 µM; the concentration of DNP (unless otherwise indicated) was 50 µM. The gray line represents the control measurement without the addition of CATR or DNP. (**C**) Quantification of MMP recovery by CATR (brown) in the presence of 30 µM DNP (yellow, brown). Data are the mean ± SD of at least three independent experiments; ** *p* < 0.01. For other conditions, see Section 2 (Materials and Methods).

**Figure 2 biomolecules-11-01178-f002:**
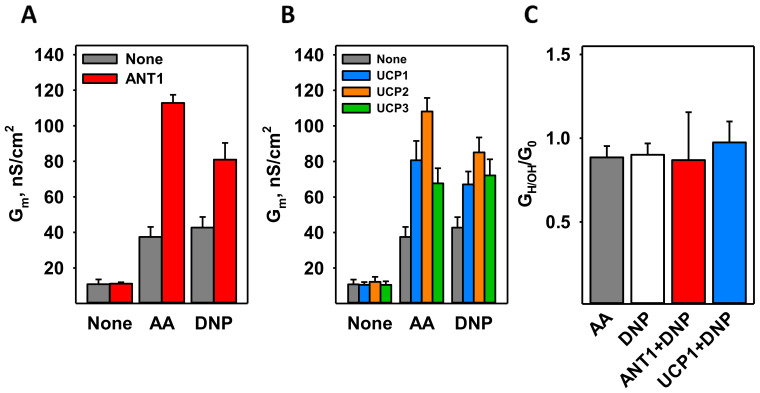
DNP-mediated proton transport in the presence of mitochondrial membrane proteins reconstituted in planar lipid bilayers. (**A**) Increase in the total specific membrane conductance (G_m_) in the presence of arachidonic acid (AA) or DNP without (gray) and with ANT1 (red). (**B**) Increase in G_m_ in the presence of AA or DNP without protein (gray) or with UCP1 (blue), UCP2 (orange), or UCP3 (green). (**C**) The ratio of proton conductance to total membrane conductance (G_H/OH_/G_0_) was measured at a transmembrane pH gradient of 0.4. The concentrations of AA and DNP were 15 mol% and 50 µM. For all measurements, the membranes were made of DOPC:DOPE:CL (45:45:10 mol%). Lipid and protein concentrations were 1.5 mg/mL and 4 µg per mg of lipid. The buffer solution consisted of 50 mM Na_2_SO_4_, 10 mM Tris, 10 mM MES, and 0.6 mM EGTA at pH = 7.34 and T = 32 °C. Data are the mean ± SD of at least three independent experiments.

**Figure 3 biomolecules-11-01178-f003:**
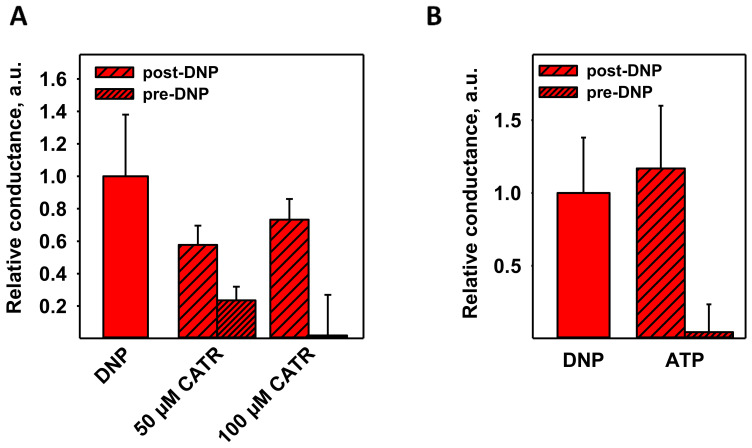
DNP-mediated proton transport in the presence of ANT1 can be inhibited by CATR or ATP. (**A**) Relative conductance of bilayer lipid membranes reconstituted with ANT1 in the presence of DNP and CATR added either after DNP (post-DNP) or before DNP (pre-DNP). (**B**) Relative conductance of bilayer membranes reconstituted with ANT1 in the presence of DNP and 4 mM ATP. Relative conductance describes the ratio between the total membrane conductance, G_m_, in the presence or absence of inhibitors to the membrane conductance measured in the presence of lipids alone (see Equation (1)). Other experimental conditions were the same as in Figure 2.

**Figure 4 biomolecules-11-01178-f004:**
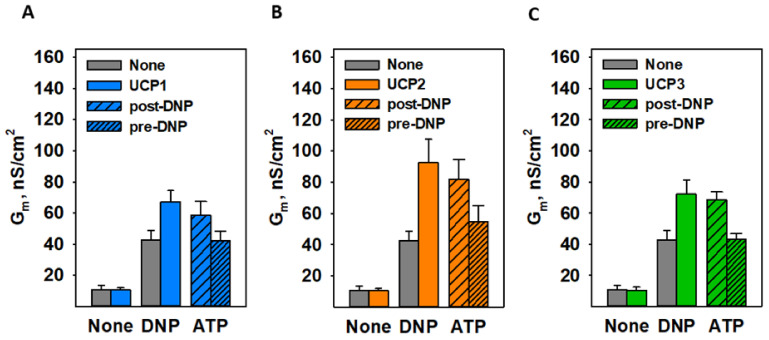
ATP decreased DNP-mediated proton transport enhanced by UCP1 (**A**), UCP2 (**B**), and UCP3 (**C**). Here, 4 mM of ATP was added either post-DNP (medium pattern) or pre-DNP (fine pattern). The experimental conditions were similar to those in Figure 2.

**Figure 5 biomolecules-11-01178-f005:**
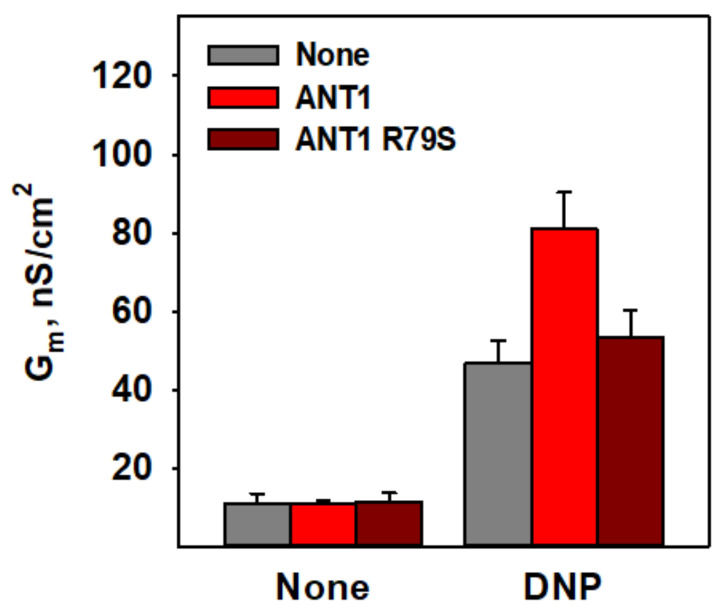
Effect of the ANT1 R79S mutation on DNP-mediated proton transport. Total specific membrane conductance (G_m_) of the lipid bilayer membranes reconstituted with ANT1 (red) or ANT1 R79S (dark red) in the presence and absence of 50 µM DNP. The experimental conditions were the same as in Figure 2.

**Figure 6 biomolecules-11-01178-f006:**
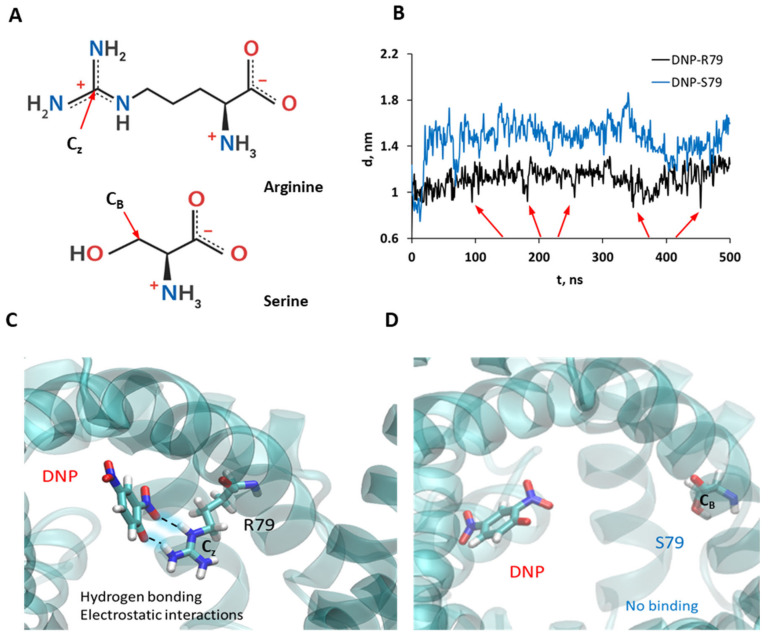
DNP binding in ANT1 and ANT1 R79S. (**A**) Structure of arginine and serine at pH = 7 in water. C_Z_ is the central carbon atom of the guanidinium side chain in R79, and C_B_ is the carbon atom in the side chain of S79. (**B**) Analysis of the distances between DNP’s center of mass and the C_Z_ atom in R79 (black) and the C_B_ atom in the S79 residue (blue). Red arrows indicate close contacts between DNP and R79. Selected snapshots from MD simulations of DNP in ANT1 (**C**) and DNP in ANT1 R79S (**D**).

## Data Availability

The datasets generated and/or analyzed during the current study can be obtained upon reasonable request from the corresponding authors or at the following link: DOI:10.5281/zenodo.5113039.

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
