# Peer review of "Mitochondrial Uncoupling Proteins (UCP1-UCP3) and Adenine Nucleotide Translocase (ANT1) Enhance the Protonophoric Action of 2,4-Dinitrophenol in Mitochondria and Planar Bilayer Membranes"

_biomolecules, 2021, doi:10.3390/biom11081178_

Round 1

Reviewer 1 Report

In this manuscript, Authors investigated the mechanism of mitochondrial uncoupling by 2,4-dinitrophenol (DNP). In particular, they showed that DNP (i) uncouples mitochondrial membrane potential in isolated mitochondria in a ANT-dependent manner, (ii) potentiates the conductance of recombinant ANT1, UCP2 and UCP3 in planar lipid bilayers and (iii) requires ANT1 R79 for its protonophoric effect. In general, the work reports a potential interesting finding which would add a small piece to the knowledge of the mechanism of action of DNP. However, I found some issues that Authors should address before this work becomes suitable for a publication.

I believe that the title is a little overstating the real contribution of the work which is most focused on the ANT respect to the UCPs, while nothing was reported for other mitochondrial proton-transporting proteins. The analysis of DNP on the UCPs appears indeed very marginal compared to that on ANT (just one set of experiments presented in Figure 2B). Authors could improve the UCP part by testing for example whether UCPs inhibitors (like GDP) can restore membrane potential upon DNP stimulation in isolated mitochondria, as done with CATR, and prevent DNP potentiation of recombinant UCPs in planar lipid bilayers. Having saying that, I still have a major concern about the choice of UCP2 and UCP3 for this study, the uncoupling activity of which is still highly debated. Authors should better test the effect of DNP on UCP1 in first place that to date represents the most accepted proton leak pathway.

Another criticism regards the electrophysiological measurements and in particular, I wonder how specific is this system to let the detection uniquely of proton transport. The use of solutions containing salts that dissociate into large anions and cations, normally impermeant through ion channels or transporters, like for example tetramethylammonium hydroxide (TMA), would allow the examination of proton currents more precisely. In alternative, can Authors test the effect of DNP on ANT and UCPs at different pH to confirm that currents are actually ascribable to proton movements? Moreover, why did Authors opt for a sodium-based instead of a potassium-based solution which should more closely mimic the intracellular ionic conditions? In general, I would also recommend to show, together with histograms, representative current traces.

Minor points:

Authors referred to a recent publication in support of the FA-stimulated proton transport by ANT (Ref 26). I believe that they should also cite the work of Bertholet et al (Nature 2019) which first showed the ANT-mediated proton currents activated by FA, with a special emphasis to the role covered by ANT1 in heart and skeletal muscle.      

In Figure 1, please provide a quantification of the extent of membrane potential uncoupling (with DNP) and recoupling (with CATR) with appropriate statistics. The (control) gray line in panel A is not shown. For all other Figures please provide an appropriate statistical analysis.  

The legend of Figure 3 is not correctly set (it appears in the main text).

In the Supplementary Fig. 1, UCP3 is not shown in the WB (or UCP2 was wrongly cited twice). Only one batch for ANT1 is also reported in the image.

Reviewer 2 Report

The study investigates the mechanism of 2,4 Dinitrophenol (DNP)-mediated proton leakage of lipid bilayers as a model for mitochondrial membranes. It is shown that the DNP stimulated conductance of the lipid bilayer increases further, when the ADP/ATP exchanger protein (ANT) is present. ATP and CATR, an inhibitor of ANT, prevent the increased conductance suggesting a competitive binding mechanism between DNP and ATP. Molecular modeling shows that R79 of the ANT is a critical residue for binding.

General comments

The study is well set-up, novel and provides a model for possible interaction between DNP and ANT. However, some small adjustments are needed before the study is ready for publication:

  1. Were controls performed with the solvents for AA and DNP (Fig. 1-4)?
  2. Significances in the plots are not shown - please add
  3. Please explain model I in the discussion for the action of FA/protein/conductance, respectively DNP/protein/conductance a little more detailed
  4. Fig. 5_ please show an empty model of the ANT around R79 and - if possible - overlay the structure with the structure when DNP R79 is bound. Else, discuss, whether a conformational change is expected due to substrate/inhibitor binding. The study would profit from modeling  ANT and ATP, too - Fig. 5a and S3 are not easy to understand

Minor:

  1. AA is often used as abbreviation for Antimycin A, an inhibitor of CIII. Maybe choose a different abbreviation.
  2. Fig. 3: A is not visible

Round 2

Reviewer 1 Report

Authors addressed all points I raised. I believe that the revision singificanlty improved the quality of the manuscritp. I do not have further comments to add.